# Monitoring Parkinson's Disease In-the-Wild

Cyrille E. Mvomo
*Department KPE, CRBLM and DART*
*McGill University, CRIR and LLUI*
Montreal, Canada and Vitznau, Switzerland
cyrille.mvomo@mail.mcgill.ca

Jordan Bedime
*Department KPE and CRBLM*
*McGill University and CRIR*
Montreal, Canada
jordan.bedime@mail.mcgill.ca

Sara Perfetto
*Department KPE and CRBLM*
*McGill University and CRIR*
Montreal, Canada
sara.perfetto@mail.mcgill.ca

Dahlia Leibovich
*Department KPE*
*McGill University*
Montreal, Canada
dahlia.leibovich@mail.mcgill.ca

Clara Guedes
*Department KPE and CRBLM*
*McGill University and CRIR*
Montreal, Canada
clara.guedes@mail.mcgill.ca

Alexandra Potvin-Desrochers
*CISSS and École Interdisciplinaire de Santé*
*Université du Québec en Outaouais*
Gatineau, Canada
alexandra.potvin-desrochers@ssss.gouv.qc.ca

Philippe C. Dixon
*Department KPE*
*McGill University*
Montreal, Canada
phil.dixon@mcgill.ca

Chris Awai Easthope
*Data Analytics and Rehabilitation Technology (DART)*
*Lake Lucerne Institute (LLUI)*
Vitznau, Switzerland
chris.awai@llui.org

Caroline Paquette
*Department KPE and CRBLM*
*McGill University and CRIR*
Montreal, Canada
caroline.paquette@mcgill.ca

*Abstract*—In Parkinson's disease (PD), the development of accurate wearable biomarkers for real-world monitoring is a priority. Developers tend to prioritize agreement with clinical features (e.g., neurological tests). However, wearable biomarkers should also reflect the pathogenic processes underlying these clinical features. This critical aspect is often overlooked in validation studies, raising doubts about construct validity and limiting adoption of these biomarkers. Here, we propose a solution to address this gap. We examined whether a previously validated wearable biomarker, derived from a deep learning model trained on raw accelerometer signals during walking to estimate motor symptom severity scores, can also reflect the pathogenic processes associated with motor dysfunction in people with PD (PwP). The model was reproduced and evaluated in-the-wild, before being deployed on a subset of PwP for whom neuroimaging data were also available. Neuroimaging data were analyzed to extract the brain activity pattern associated with predicted motor symptoms severity scores. The topographic similarity between the extracted pattern and two established brain patterns (one underlying motor symptoms in PD and one not) was assessed. The model accurately estimated ground-truth motor severity scores (mean absolute error = 5.20). Despite not being explicitly trained for this purpose, the model was also able to capture pathogenic mechanisms specifically linked to motor dysfunction in PD (dice similarity = 0.653). These findings represent an initial step toward linking wearable biomarkers not only to clinical features, but also to underlying mechanistic representations. This supports the wider adoption of wearable biomarkers in clinical practice and trials.

*Keywords*—*wearable biomarkers; Parkinson's disease; real-world monitoring; neuroimaging; deep learning.*

This work was supported by *Parkinson Canada*, the *Center for Research on Brain, Language and Music at McGill University (CRBLM, Faculty of Medicine and Health Sciences)*, the *Natural Science and Engineering Research Council of Canada*, the *Quebec BioImaging Network* and the *Canada Foundation for Innovation*. The resources to reproduce this work are on GitHub at *https://github.com/cyrillemvomo/Sensing_Parkinson_In_The_Wild*

## I. INTRODUCTION

Parkinson's disease (PD) is among the fastest-growing neurological conditions worldwide [1]. It manifests through a complex spectrum of motor and non-motor symptoms [1], [2] that emerge from disruptions across multiple neurotransmitter systems and as side effects of pharmacological interventions [1]. The slow progressive and disabling nature of these symptoms demands sustained monitoring. Current options for symptom monitoring include traditional biomarkers and clinical features, each with its own limitations. Traditional biomarkers (e.g., neuroimaging [3] or fluids [4]) are invasive, costly, or require specialized personnel and infrastructure. Clinical features – such as the gold standard Movement Disorder Society Unified Parkinson's Disease Rating Scale (MDS-UPDRS) [5]– are semi-subjective [6], long to administer (~20 minutes) [5], and rely on episodic evaluations conducted in clinics by increasingly scarce movement disorder specialists [7]–[9]. Given these limitations, there is a pressing need for alternatives. Advances in wearable sensing technologies and data analytics have fueled the development of wearable biomarkers [7], [10]–[13]. These tools promise to overcome current limitations by enabling objective, continuous, non-invasive, and cost-effective assessments in real-world environments [7], [10]–[13]. Recent years have therefore witnessed growing efforts to establish their validity across several domains (e.g., technical, etc.) [14]–[16]. However, doubts remain about construct validity [14], [17].

Much of this skepticism stems from a fundamental design gap in validation studies: researchers tend to solely focus on optimizing performance metrics, such as agreement with a clinical feature [7], [11]–[13], [18]–[21]. Yet, beyond agreeing with a clinical feature, a wearable biomarker must, by definition, reflect the underlying pathogenic processes associated with that

feature [22], [23]. Nonetheless, this aspect is largely overlooked in validation studies and raises a *what do we measure* question (Fig. 1). Addressing this question is urgently needed for several reasons. First, wearable biomarkers are inherently influenced by multiple sources of variability and also capture traits unrelated to disease-induced behavioral decline [24], especially when employed in unsupervised environments (e.g., home). Second, their ability to capture pathogenic changes associated with clinical features has been identified as a potential barrier to their further adoption [17]. And most importantly, these tools are already being used despite this lack of clarity [25], [26], which could negatively impact decision-making.

Here, we present an attempt to answer this question by reassessing a previously validated wearable biomarker (referred to here as "*wearable-based* model") successfully developed to predict motor severity scores of the MDS-UPDRS (MDS-UPDRS III) from raw accelerometer data (mean absolute error = 6.29) [27]. Our goal was to evaluate whether, beyond accurately predicting ground-truth MDS-UPDRS III scores, the *wearable-based* model also captures the pathogenic processes underlying motor symptoms in PD.

The present paper is organized as follows. Section II describes the dataset, the procedure used to reproduce and evaluate the *wearable-based* model, and the method applied to assess its ability to capture the pathogenic processes underlying motor symptoms in PD. Section III presents and discusses the results. Section IV outlines the limitations and presents directions. Section V concludes the paper. Finally, Section VI highlights the significance of the contribution.

## II. METHODS

### A. Dataset

This study combines previously acquired data from four distinct sources: ONPAR [28], DeFOG [29], LRS [30], and BRAIN-PET [31]–[33] (Fig. 2). Across all sources, people with Parkinson's disease (PwP) were selected to ensure a balanced distribution of MDS-UPDRS III scores.

ONPAR was an observational study investigating the impact of disease severity on brain activity [28]. It included 19 PwP, who were assessed in home settings using AX3 and AX6 sensors (Axivity Ltd., York, UK; 23 × 32.5 × 7.6 mm; 11 g; 100 Hz sampling rate). No neuroimaging data were collected.

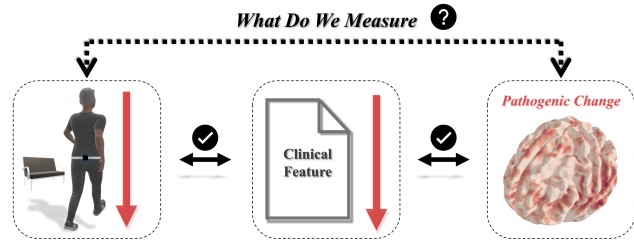

Fig. 1. Schematic of the *what do we measure* question. A PwP walking in-the-wild while wearing a lower-back sensor. Data from this sensor are used to generate the output of an already validated wearable biomarker, which reflects behavioral decline (red arrows) and agrees with a clinical feature. This clinical feature is, in turn, known to be related to pathogenic changes in the brain. The central question is whether the output of the wearable biomarker is also associated with these pathogenic changes.

DeFOG was a randomized controlled trial evaluating a personalized cueing system for gait impairments [29]. It included 45 PwP who did not receive cueing. Participants were assessed at home using the same AX3 and AX6 sensors, without neuroimaging.

LRS was an observational study assessing the ability of wearable technologies to quantify symptom severity [30]. It included 6 PwP, who were also assessed at home without neuroimaging, using Shimmer3 sensors (Shimmer Research Ltd., Dublin, Ireland; 51 × 34 × 14 mm; 23.6 g; 50 Hz sampling rate).

BRAIN-PET included 23 participants from several observational studies on brain activity during walking: 16 PwP and 7 healthy controls (HC) [31]–[33]. Participants underwent a laboratory-based "steering" protocol using Opal V2R sensors (Clario, Philadelphia, USA; 55 × 40.2 × 12.5 mm; 23.6 g; 128 Hz sampling rate). Neuroimaging data (T1-weighted MRI and 18F-FDG PET) were acquired simultaneously to capture both brain anatomy (Siemens Prisma 3-T MRI) and metabolic activity (CTI/Siemens HRRT PET). The steering protocol was designed to simulate real-world ambulatory challenges by incorporating continuous turns, which account for nearly half of daily steps [34] and are especially relevant in PD [35].

Participants from ONPAR, DeFOG and LRS constituted the *home* cohort, while BRAIN-PET participants formed the *laboratory* cohort. Clinical and demographic information was available for all participants. Ethical approval was obtained from

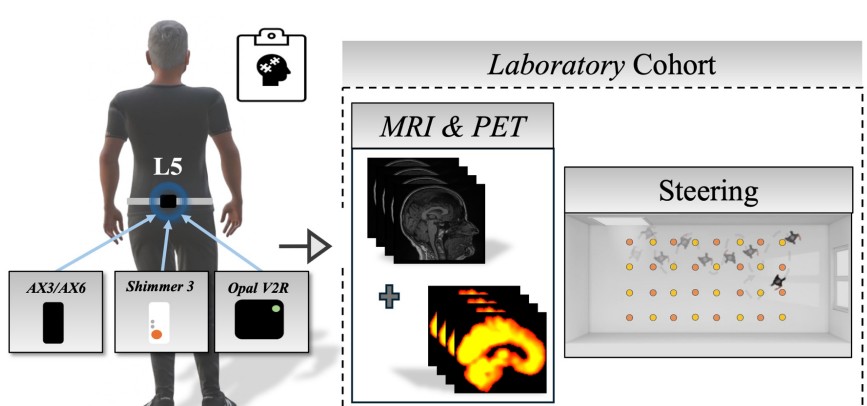 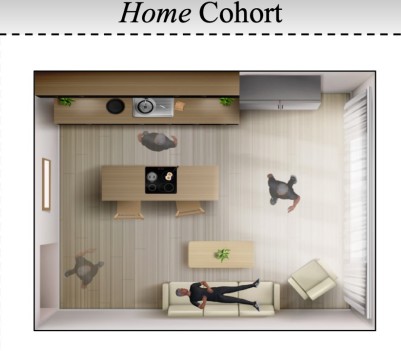

Fig. 2. Overview of the dataset.

the appropriate institutional review boards, and written informed consent was provided by all participants. Additional details regarding eligibility criteria and study protocols are available elsewhere [28]–[33].

## B. Wearable Data Preprocessing

For the *home* cohort, wearable data underwent minimal preprocessing. Gait sequences longer than 30 seconds (reflecting purposeful walking episodes) were identified using a validated algorithm [36] previously used in PD research [11]. Only sequences occurring during the first two days were retained. This duration has been shown to be sufficient to reliably capture real-world locomotor behavior in PD (test-retest reliability > 85%) [37].

For the *laboratory* cohort, although participants were recorded for 20 to 30 minutes to meet radiotracer uptake requirements, only data from the first 2 minutes were used. This approach ensured the capture of true ambulatory performance by aligning with established real-world gait bout lengths in PD [38]. Since predefined walking protocols ensured gait sequences, no additional gait sequence detection was necessary for the *laboratory* cohort.

The remaining preprocessing steps were consistent across all cohorts. Each gait sequence was divided into continuous, non-overlapping 5-second sliding windows [27]. From each window, a signal magnitude vector $a_{mag}$ was computed by combining the three components of the raw triaxial accelerometer signal $a(t)$ using the following formula:

$$a_{mag} = \parallel a(t) \parallel_2 = \sqrt{a_{vertical}^2 + a_{ML}^2 + a_{AP}^2} \qquad (1)$$

To ensure consistency, signals from all $a_{mag}$ windows were resampled to 100 Hz (i.e., 1 $a_{mag}$ window = 500 samples). No additional preprocessing was applied.

## C. Wearable-Based Model

The *wearable-based* model corresponded to the one validated by Rehman et al. and was replicated in-the-wild (Fig. 3) [27]. It consisted of a Deep Convolutional Neural Network Regressor designed to predict MDS-UPDRS III scores from 5-second $a_{mag}$ windows (ON-medication). Although Rehman et al. did not validate the *wearable-based* model for real-world use, it was selected for three main reasons: (1) the clinical relevance of the targeted problem, (2) its promising performance for monitoring motor symptoms longitudinally despite being trained on a relatively small dataset, and (3) its potential for clinical translation in-the-wild, as it relies solely on raw acceleration signals from a single sensor that can be passively and continuously collected using widely accessible and easy-to-

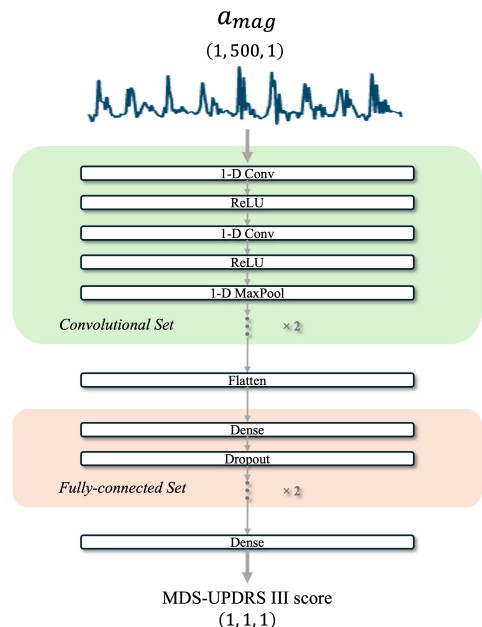

Fig. 3. *Wearable-based* model architecture (adapted from Rehman et al.).

wear devices (e.g., smartphones), without requiring feature engineering or participant input.

Briefly, the *wearable-based* model comprises two main components. The first is a set of convolutional layers designed to extract deep spatial features from each $a_{mag}$ window. The second component includes fully connected layers that deep-learned the relationship between the extracted features and the target MDS-UPDRS III scores.

The *wearable-based* model was constructed with the dual goal of ensuring real-world relevance and fidelity to the original implementation. Accordingly, training, evaluation, and testing were performed exclusively on the *home* cohort. The mean absolute error (MAE) was chosen to measure agreement with ground-truth MDS-UPDRS III. *Home* participants were assigned to training (~70%), validation (~10%), and test (~20%) sets (regardless of data source or sensor type) using an inter-subject split scheme [39]. To preserve fidelity to the original approach, hyperparameter tuning was limited to the subset of hyperparameters previously optimized by Rehman et al. (see [27]) and was conducted using grid search solely on the validation set. Training employed early stopping with a patience of 10 epochs to prevent overfitting.

## D. Image Processing

Once acquired from *laboratory* participants, brain scans were processed using previously described methods (Fig. 4)

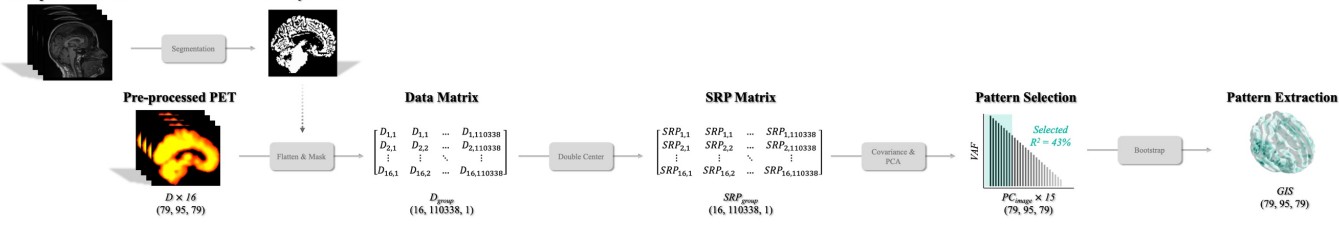

Fig. 4. Pattern identification procedure. VAF denotes variance accounted for.

[33]. Briefly, this included: (i) co-registering PET to MRI images; (ii) spatially normalizing both scans to the Montreal Neurological Institute (MNI) space; (iii) smoothing PET images using a Gaussian filter (8 mm in full width at half maximum); and (iv) segmenting MRI scans.

### E. Pattern Identification

Following previous methodologies [40]–[42], a principal component analysis-based algorithm incorporating scaled subprofile modeling (SSM/PCA) was applied to extract brain activity patterns associated with MDS-UPDRS III scores predicted by the *wearable-based* model during walking in PwP from the *laboratory* cohort (Fig. 4).

Each PwP scan $D$ was first flattened and masked (using an age-specific binary mask derived from MRI segmentations) to construct a group-level data matrix $D_{group}$ with 16 rows (one per PwP) and 110,338 columns (voxels). $D_{group}$ was double-centered to obtain the subject residual profile matrix $SRP_{group}$, from which a covariance matrix was computed. PCA was then applied to extract Principal Component image patterns ($PC_{image}$), each yielding an expression score per participant. $PC_{image}$ or sets of $PC_{image}$ significantly associated with the predicted MDS-UPDRS III scores were identified through regression (P < 0.05; 1,000 permutations). Model selection was based on the lowest Akaike Information Criterion. Z-scored group-invariant subprofile (GIS) pattern maps were generated using 1,000 bootstrap resamples (for correction), and brain regions contributing significantly to the GIS pattern were identified using the AAL3.1 atlas ($|z| \geq 1.64$; P < 0.05; one-tailed). More details about SSM/PCA are available elsewhere [40]–[42].

### F. Pattern Analysis

The ability of the GIS pattern to reflect brain mechanisms underlying disease-specific motor dysfunction was evaluated in participants from the *laboratory* cohort. Initially, its disease specificity was determined by comparing GIS expression scores between PwP and HCs using ANCOVA (P < 0.05). For this purpose, HC expression scores were prospectively computed on a single-case basis using a voxel-based automated algorithm, and all expression scores (PwP and HC) were standardized using the HC group as reference [40]–[42].

Two key aspects were then evaluated: (i) the topographic similarity between the GIS pattern and the PD-related motor pattern (PDRP); (ii) the topographic similarity between the GIS pattern and the normal motor-related pattern (NMRP). The PDRP is a widely established brain network that is specific to PD and uniquely reflects disease-related motor progression [43], [44], while the NMRP is a network known to reflect brain activity during normal movement and is not specific to motor symptoms in PD [45]. The topography of both the PDRP and NMRP has been previously described in the literature [43], [45]. To evaluate the similarity, significant regions in the GIS, PDRP, and NMRP patterns were converted into binary masks. Topographic similarity between the GIS mask and each of the two reference pattern masks (PDRP and NMRP) was assessed using the Dice Similarity Coefficient (DSC), with a threshold set at 0.5. The goal was to determine whether the GIS pattern (associated with predictions from the *wearable-based* model) exhibited significant similarity with the PDRP, while remaining dissimilar to the NMRP, thereby demonstrating the capacity of the GIS pattern (and thus the *wearable-based* model) to specifically and uniquely capture pathogenic processes of motor symptoms in PwP.

### III. RESULTS AND DISCUSSION

Participant characteristics are detailed in Table 1. No participants were excluded from either cohort. The training set included PwP with a broader distribution of age, disease duration, and motor severity compared to PwP in the validation and test sets, as well as in the laboratory cohort. This diversity in participant characteristics supports the relevance of the training set in promoting model generalizability. Approximately 3,000 $a_{mag}$ windows were available for each participant in the home cohort, compared to ~24 for those in the laboratory cohort.

### A. Predicting Ground-Truth MDS-UPDRS III

The final set of tuned hyperparameters is provided in Table 2. The model was evaluated on the test set. For each participant, predictions were first averaged to obtain a single estimated score. The MAE was then computed as the average absolute difference between these estimates and the corresponding ground-truth MDS-UPDRS III scores across all participants [27]. Although the wearable-based model showed higher errors at the more extreme score ranges (Fig. 5a), it achieved an overall MAE of 5.20 (95% CI [3.14, 7.69]) in predicting ground-truth MDS-UPDRS III scores. This performance closely aligned with the original results reported by Rehman et al. (Fig. 5b) [27]. This supports the ability of the wearable-based model to estimate MDS-UPDRS III scores in ecological settings.

### B. Similarity with PDRP and NMRP

The extracted GIS pattern is presented in Fig. 6. The analysis was restricted to the first 8 $PC_{image}$ (~85% VAF). In laboratory PwP, the MDS-UPDRS III scores estimated by the wearable-based model were also close to ground truth (MAE = 4.83; 95% CI [3.02, 6.98]) and were best explained by a linear combination of $PC_{image}$ 3, 7, and 8 ($R^2 = 65\%$; 1-β = 0.99; P = 0.001). Significant clusters were found in various subcortical regions, including the basal ganglia, brainstem, and cerebellum,

TABLE I.    PARTICIPANTS CHARACTERISTICS

| Variable (unit/range) | Home | | | Laboratory | |
|---|---|---|---|---|---|
| | Train | Val. | Test | PwP | HC |
| N | 51 | 6 | 13 | 16 | 7 |
| Age (y) | 65 ± 7 (51-82)[a] | 71 ± 1 (69-72)[a] | 69 ± 8 (54-83)[a] | 64 ± 5 (54-72)[a] | 60 ± 2 (58-63)[a] |
| Sex (% F) | 37% | 31% | 50% | 38% | 57% |
| PD Duration (y since diagnosis) | 9 ± 12 (1-30)[b] | 11 ± 4 (4-16)[a] | 8 ± 3 (5-13)[a] | 7 ± 3 (1-13)[a] | NA |
| MDS-UPDRS III ON (0-132) | 30 ± 11 (5-56)[a] | 28 ± 8 (13-35)[a] | 34 ± 5 (26-44)[a] | 33 ± 5 (22-40)[a] | NA |

NA : non-applicable. [a]Mean ± SD (min-max). [b]Median ± IQR (min-max).

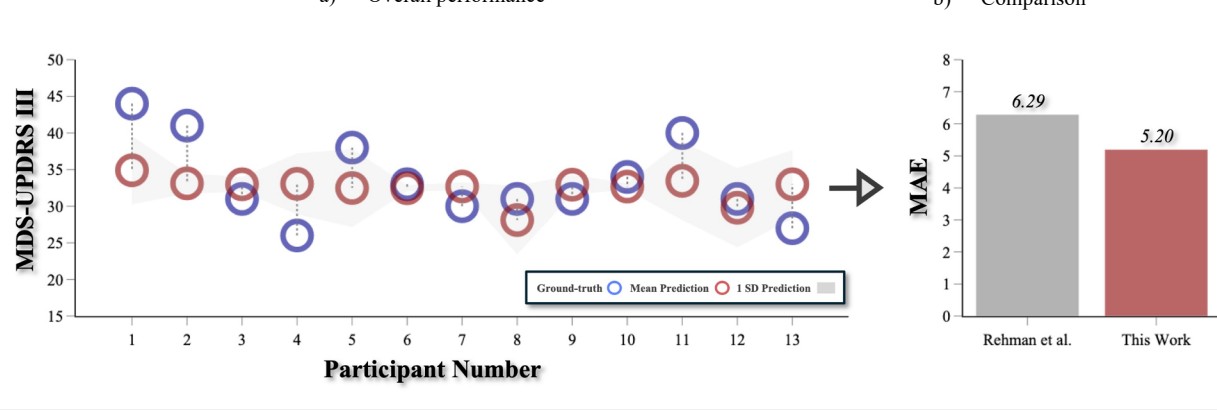

a) Overall performance

b) Comparison

Fig. 5. *Wearable-based* model performance.

a) GIS Topography

b) Comparison with HC

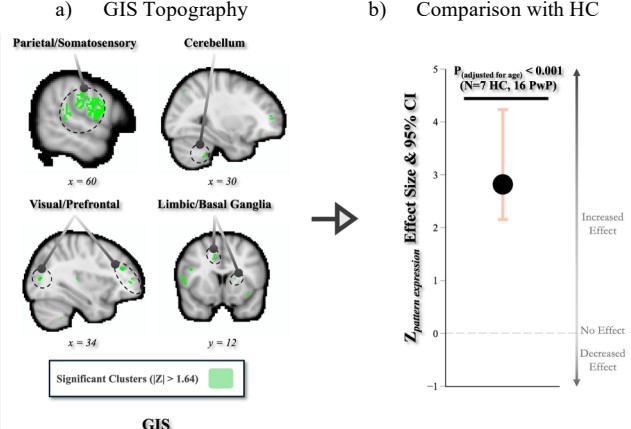

Fig. 6. GIS pattern. a) Clusters examples. b) Forest plot of the difference in GIS expression between PwP and HC.

TABLE II.    TUNED HYPERPARAMETERS

| Hyperparameter | | | | | | | |
|---|---|---|---|---|---|---|---|
| *conv1_out* | *conv2_out* | *Kernel Size* | *Fc units* | *Drop. Rate* | *Lear. Rate* | *Batch Size* | *Epoch* |
| 8 | 8 | 3 | 100 | 0 | 0.001 | 128 | 50 |

as well as in primary visual, somatosensory, unimodal and heteromodal cortices, and limbic regions (Fig. 6a). GIS pattern expression was PD-specific (P < 0.001; Fig. 6b). The GIS mask exhibited no spatial similarity with the NMRP (DSC = 0.402; 95% CI [0.399, 0.405]; Fig. 7a, 7b and 7c) and significant similarity with the PDRP, although the similarity with the PDRP remained moderate (DSC = 0.653; 95% CI [0.650, 0.655]). These findings suggest that, despite not being explicitly trained for that purpose, the *wearable-based* model

not only accurately predicts ground-truth MDS-UPDRS III scores but also captures underlying pathogenic processes specifically related to motor symptoms in PD.

## IV. LIMITATIONS AND NEXT STEPS

Some limitations should be acknowledged. First, although the model performed well overall, prediction errors were larger in individuals with more severe motor symptoms. This trend, also observed in related studies [13], suggests the model may not yet be suitable for deployment across the full spectrum of motor severity. However, assessing clinical deployability is beyond the scope of the present work as the aim of this work was rather to faithfully reproduce how the model validated by Rehman et al. would perform in real-world settings and explore its potential to capture pathogenic processes [27]. Second, although linking wearable biomarkers to pathogenic mechanisms is crucial for improving construct validity, the task-based 18F-FDG PET protocol used in BRAIN-PET is

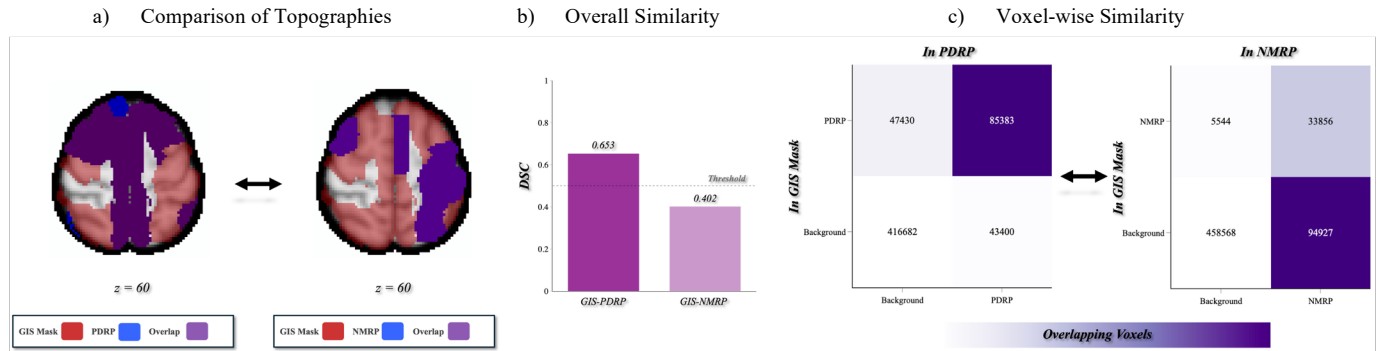

a) Comparison of Topographies

b) Overall Similarity

c) Voxel-wise Similarity

Fig. 7. Pattern similarity. a) Overlay example of GIS-PDRP and GIS-NMRP masks. b) Dice similarity. c) Voxel-wise confusion matrices showing overlapping voxels between GIS-PDRP and GIS-NMRP, controlling for differences in pattern size. Background voxels are excluded from the colorscale.

limited. Specifically, the prolonged laboratory walking required during tracer uptake does not fully reflect real-world scenarios (~20 minutes of continuous turning) [31]–[33]. We mitigated this by restricting gait data to established real-world walking bout lengths in PD (although the images still reflect 20–30 minutes of gait) [38], but future work should explore imaging modalities more representative of real-life behavior.

Our further investigations will focus on refining the *wearable-based* model to enhance its generalizability. Additionally, we will explore how "*learning the pathogenic processes*" and accounting for variability across medication states could further improve model performance. At that stage, it will be necessary to longitudinally assess compliance, robustness across sensor modalities and locations, and sensitivity to minimal clinically important differences.

## V. CONCLUSION

Wearable biomarkers promise to transform PD monitoring, but doubts regarding their construct validity continue to limit their adoption [14], [17]. To address these doubts, a shift toward establishing "explainable" digital phenotypes has been proposed [46]. This paradigm aims to bridge the disconnect between end-users (e.g., clinicians) and developers [47] by involving multiple levels of biomarker validation, including neurobiological validation beyond clinical validation alone. In this work, we examined whether a previously validated *wearable-based* model, designed to estimate MDS-UPDRS III scores, can also reflect the pathogenic processes driving motor dysfunction in PD. Our findings suggest that, in addition to providing accurate estimates of motor severity in real-world environments, the model also specifically captures brain processes known to underlie PD motor symptoms. Therefore, this work marks an initial step toward establishing explainable digital phenotypes and supports the broader integration of wearable biomarkers into both clinical practice and research, where they could aid in monitoring symptom progression or detecting treatment effects [14], [48].

## VI. SIGNIFICANCE OF THE CONTRIBUTION

Methodologically, this work is the first to combine deep learning, wearable sensing in ecologically valid settings, and task-based functional neuroimaging to evaluate the validity of wearable biomarkers in PD. In doing so, it introduces a multifactorial approach that aligns with the widely acknowledged complexity of PD and its behavioral manifestations. As such, this study may represent a methodological shift in the validation of wearable biomarkers and could serve as a reference for future investigations in the field. In addition, it is the first to employ the promising SSM/PCA methodology to extract disease-related brain activity patterns from task-based neuroimaging paradigms, thereby proposing a novel framework for clinical informatics.

Its most impactful contribution, however, lies in its clinical dimension. It transcends traditional disciplinary boundaries by being among the first to show that, beyond agreement with clinical features, a wearable biomarker can capture "true"

disease-induced neural signatures underlying those features in PD. In doing so, it directly addresses concerns of "data fundamentalism" [49], a critique frequently directed at wearable biomarkers whose ability to reflect underlying pathophysiology has been questioned [17], [46], [49]. By demonstrating that wearable biomarkers can index disease-related neural processes, this work reinforces their legitimacy as true biomarkers consistent with the fundamental definition of a biomarker. Clarifying the neurobiological validity of wearable biomarkers has implications for clinical adoption. Overcoming skepticism among PD specialists could accelerate their integration into clinical research and practice. Neurobiologically informed wearable biomarkers could improve participant selection in trials by offering a practical way to identify individuals whose profiles align best with the target intervention, enable objective and rapid detection of response both in trials and routine, and facilitate remote monitoring to reduce attrition and non-adherence. These advances could shorten the duration and cost of disease- and symptom-modifying interventions, thereby accelerating the discovery of effective therapies.

## ACKNOWLEDGMENT

We thank the *participants*, the *MNI*, the *McGill Ice Hockey Group*, *Mobilise-D* (and collaborators) for their contributions.

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
