# OpenReview forum: "Monitoring Parkinson’s Disease In-the-Wild"
_IEEE.org/EMBS/BHI/2025/Conference — BHI 2025_

### Official Review · Reviewer_zqjQ · 2025-07-14
**This manuscript reproduces and extends a previously published deep-learning model that converts raw lower-back accelerometer windows into MDS-UPDRS III motor-severity scores. In 70 “home” participants with two days of unsupervised gait data, the model achieved a mean-absolute-error of 5.2. In a separate “laboratory” cohort the same model (MAE 4.8) was paired with FDG-PET/MRI; a scaled-subprofile-modelling analysis showed that the biomarker’s predicted scores express a disease-specific metabolic pattern overlapping the canonical Parkinson’s-disease–related pattern (PDRP, Dice 65 %), but not the normal motor pattern (NMRP, Dice 40 %). The study argues that a validated wearable biomarker can also capture underlying pathogenic processes, thereby strengthening its construct validity and potential clinical utility.**

**Confidence:** 5
**Clarity Of Writing:** great
**Clinical Significance:** good
**Methodological Novelty:** great
**Overall Rating:** 8

**Experiments And Results:**

great

**Questions For The Authors:**

1. Can you report performance on a sensor modality or body location not seen during training to demonstrate robustness to hardware variance?
2. How does the wearable biomarker compare with established commercial systems (e.g., PKG, STAT-ON) in terms of MAE and user burden?
3. Did you explore attention or saliency within the CNN to identify which gait-cycle segments drive high predicted scores?
4. Given that MDS-UPDRS fluctuates with medication, were ON/OFF states balanced across participants, and could state-specific models perform better?
5. What power calculation supports the n = 16 PET sample as sufficient for SSM/PCA; how stable is the GIS when bootstrapping participants?
6. Have you considered validating against dopamine-transporter SPECT or neuromelanin MRI, which may better capture nigrostriatal degeneration than FDG metabolism?

**Strengths:**

1. Demonstrates that a wearable gait biomarker trained for clinical agreement also reflects disease-specific neural metabolism, addressing long-standing doubts about construct validity of digital endpoints.
2. Bridges real-world sensing, deep-learning analytics, and neuroimaging, offering a multifactorial validation pipeline aligned with calls for “explainable digital phenotypes”.
3. Uses unsupervised, ecologically valid data collected with commodity inertial sensors—consistent with emerging cohorts that monitor PD progression outside the clinic.
4. Employs SSM/PCA to relate wearable predictions to recognised brain patterns, a technique previously validated for PDRP quantification.
5. Reports inter-subject data splits and early stopping, echoing accepted machine-learning practice to reduce over-fitting.
6. Addresses a high-impact care gap—objective, continuous assessment of motor status—highlighted by regulators and clinical-trial designers.

**Summary Of The Paper:**

Raw tri-axial acceleration from 5 s windows during purposeful walking was resampled to 100 Hz, combined into a signal-magnitude vector, and fed into a convolutional regressor whose hyper-parameters were tuned on a validation subset of the “home” cohort. Predictions per subject were averaged to yield a single estimated MDS-UPDRS III score and compared with clinician-rated ground truth (home MAE 5.20). FDG-PET scans from 16 additional patients were analysed with SSM/PCA; expression of the resulting group-invariant subprofile (GIS) explained 65 % of the variance in predicted scores and showed significant overlap with the PDRP. Healthy controls (n = 7) exhibited markedly lower GIS expression.

**Weaknesses:**

1. External validity is limited: all training data come from three convenience datasets using similar device placements; generalisability to wrist-based or smartphone sensors is untested.
2. MAE increases at higher MDS-UPDRS III scores, suggesting reduced accuracy in advanced disease—yet severe cases are those most requiring remote monitoring.
3. No direct comparison with simpler hand-crafted features or with commercial devices (e.g., PKG), leaving cost-benefit unclear.
4. PET analysis uses only 2 min of a prolonged turning task, raising concerns about representativeness of the extracted metabolic pattern.
5. The sample with neuroimaging is small (n = 16 PwP), limiting power to detect moderating effects of medication state or disease duration.
6. Reliance on clinician-scored MDS-UPDRS ground truth inherits its known rater variability and day-to-day fluctuation.
7. Confounding factors such as comorbid arthritis or cognitive load during walking are not modelled, yet can influence gait and FDG uptake.
8. The study stops short of demonstrating longitudinal sensitivity; without progression data, clinical-trial utility remains speculative.

---

### Official Review · Reviewer_XKLJ · 2025-07-18
**Multimodal validation shows promise but requires methodological strengthening before acceptance**

**Confidence:** 4
**Clarity Of Writing:** excellent
**Clinical Significance:** excellent
**Methodological Novelty:** great
**Overall Rating:** 7
**Final Rating:** 8

**Experiments And Results:**

good

**Questions For The Authors:**

What power analysis justified the n=16 neuroimaging sample? Can you provide post-hoc power calculations and confidence intervals for the 65% PDRP similarity finding?

How should clinicians interpret a MAE of 5.20 on the MDS-UPDRS III scale? What constitutes a clinically meaningful difference for monitoring purposes?

Would the neuroimaging findings remain significant after multiple comparisons correction (e.g., FDR)? How sensitive are results to the 50% Dice threshold choice?

Will code and trained model weights be made available? Can you provide more detailed preprocessing specifications for replication?

**Strengths:**

This is the first study to bridge wearable biomarkers with neuroimaging validation, addressing the critical "what do we measure" question that limits clinical adoption of digital biomarkers. The multimodal approach represents a meaningful methodological advance.

Tackles a genuine barrier to digital biomarker implementation—the need to demonstrate construct validity beyond clinical correlation. The real-world data collection across multiple sites enhances ecological validity.

The 65% PDRP similarity provides compelling evidence that data-driven models can capture disease-specific brain networks without explicit training, supporting the biological plausibility of wearable biomarkers.

Appropriate use of inter-subject train/validation/test splits, established SSM/PCA methodology for pattern extraction, and early stopping to prevent overfitting.

**Summary Of The Paper:**

This study addresses the construct validity gap in wearable biomarkers for Parkinson's disease by examining whether a deep learning model trained on accelerometer data can capture underlying pathogenic brain processes beyond clinical correlations. The authors reproduced a previously validated CNN model that predicts MDS-UPDRS III scores from 5-second accelerometer windows during walking. The model was evaluated on a home cohort (n=70 PwP across ONPAR, DeFOG, and LRS studies) and then applied to a laboratory cohort with simultaneous 18F-FDG PET neuroimaging (n=16 PwP, 7 HC from BRAIN-PET). Using SSM/PCA analysis, the authors extracted brain activity patterns associated with the model's predictions and assessed topographic similarity to established PD-related motor patterns (PDRP) and normal motor-related patterns (NMRP). The model achieved MAE=5.20 for clinical prediction and demonstrated 65% Dice similarity with PDRP while showing only 40% similarity with NMRP, suggesting capture of disease-specific pathogenic mechanisms.

**Weaknesses:**

Missing confidence intervals for key metrics (MAE, Dice coefficients), no multiple comparisons correction for neuroimaging analyses, and lack of mixed-effects modeling to account for repeated measures within subjects.

No code or data availability mentioned, incomplete methodological details for replication, and heterogeneous sensor types across cohorts without standardization protocols.

---

### Official Review · Reviewer_KnNF · 2025-07-18
**Parkinson monitoring in-the-wild**

**Confidence:** 3
**Clarity Of Writing:** great
**Clinical Significance:** good
**Methodological Novelty:** great
**Overall Rating:** 6
**Final Rating:** 8

**Experiments And Results:**

good

**Questions For The Authors:**

1.	How might model performance vary across medication states (ON vs. OFF)?
The study uses ON-medication MDS-UPDRS scores, but PD symptoms are known to fluctuate. Clarifying whether the model is robust to these fluctuations could improve its clinical interpretability. An answer indicating plans or results regarding OFF-medication validation could improve the perceived completeness of the study.
2.	Could you elaborate on why only the first two minutes of lab data were used?
The rationale seems to center on avoiding learning/fatigue effects. However, this significantly reduces the data volume. If longer recordings were available, what was the impact of excluding them? Understanding this tradeoff may change the perceived generalizability of the findings.
3.	To what extent do the differences in sensor hardware and sampling rates across cohorts affect model performance or interpretability?
The study uses data from Axivity, Shimmer3, and Opal sensors, with different sampling rates. While signals are resampled, sensor variability might introduce bias. Clarifying this could improve confidence in the generalizability of the approach.
4.	Is the GIS pattern specific only to PD motor dysfunction, or could it generalize to other movement disorders?
The authors demonstrate dissimilarity with NMRP and similarity with PDRP, but it remains unclear how disease-specific the GIS pattern is beyond PD. Discussing potential specificity could greatly strengthen the argument for mechanistic validity.
5.	Could combining wearable data and neuroimaging in a joint deep learning framework improve both prediction and biological specificity?
The study validates predictions post hoc using neuroimaging, but integration within a single multimodal model might yield better performance or insight. A response could indicate the feasibility or early results from ongoing efforts.

**Strengths:**

The following strenghts arise from the study:
•	Innovative Validation Approach: The work goes beyond traditional clinical validation by incorporating neurobiological validation, addressing a critical limitation in current wearable biomarker studies.
•	Multimodal Data Integration: By combining wearable sensor data and neuroimaging, the authors provide a robust framework for exploring construct validity in digital health.
•	Ecological Relevance: Real-world deployment of the wearable-based model enhances its translational potential for clinical use outside of controlled lab environments.
•	Explainability and Mechanistic Insight: The use of SSM/PCA to derive mechanistically meaningful patterns advances the development of "explainable" digital phenotypes.
•	Solid Reproducibility Strategy: The authors faithfully replicate an existing model (Rehman et al., 2021) with attention to methodological transparency and comparability.

**Summary Of The Paper:**

This paper investigates whether a previously validated wearable-based deep learning model—originally developed to estimate MDS-UPDRS III motor severity scores from raw accelerometer data—can also capture the pathogenic brain mechanisms underlying motor dysfunction in Parkinson’s Disease (PD). The authors reproduced and deployed the model in real-world ("in-the-wild") settings using data from multiple cohorts, including one with simultaneous neuroimaging (BRAIN-PET). The predicted symptom scores were then linked to FDG-PET-derived brain activity patterns using a scaled subprofile modeling approach (SSM/PCA). The authors assessed the similarity between the identified neural patterns and two established reference patterns: the Parkinson’s Disease-Related Pattern (PDRP) and a Normal Motor-Related Pattern (NMRP). The model demonstrated high predictive accuracy (Mean Absolute Error ≈ 5.20) and a moderate Dice similarity (65%) with the PDRP, supporting the notion that wearable biomarkers can reflect both clinical symptoms and underlying pathophysiology

**Weaknesses:**

However, several weaknesses reveal uncertainty about the quality of the work:
•	Limited Generalizability in Severe Cases: The model exhibits higher prediction error for individuals with more severe motor symptoms. This limitation could impair applicability across the full PD spectrum and warrants targeted fine-tuning.
•	Small Imaging Cohort Size: The neuroimaging dataset includes only 16 PwP and 7 controls, which limits statistical power and the robustness of GIS pattern generalization.
•	Lab-Based Imaging Protocol: Although the study uses an ambulatory-like protocol, it remains lab-constrained and may not fully reflect everyday motor behavior. PET acquisitions are inherently constrained by the need for radiotracer uptake time and may not capture rapid or dynamic neural changes.
•	Moderate Similarity with PDRP: While the Dice similarity with PDRP is meaningful, it remains modest (65%), suggesting that additional pathogenic mechanisms may not be fully captured by the current model or neuroimaging approach.
•	Unclear Longitudinal Utility: The study focuses on static validation, but it is unclear how well the model would perform in tracking disease progression or treatment response longitudinally.

---

### Official Review · Reviewer_yVNU · 2025-07-22
**Monitoring Parkinson’s Disease In-the-Wild**

**Confidence:** 5
**Clarity Of Writing:** great
**Clinical Significance:** great
**Methodological Novelty:** great
**Overall Rating:** 6

**Experiments And Results:**

great

**Questions For The Authors:**

Innovative part should be more emphasized.
Establishing explainable digital phenotypes and supports the broader integration of wearable biomarkers into both clinical practice and research should be more emphasized.

**Strengths:**

Manuscript combines deep learning, wearable sensing in ecologically valid settings, and task-based
functional neuroimaging to evaluate the validity of wearable biomarkers in PD.
It is proposing a novel automated framework for imaging and clinical informatics and clinical beneftis are very important.

**Summary Of The Paper:**

Manuscript combines deep learning, wearable sensing in ecologically valid settings, and task-based
functional neuroimaging to evaluate the validity of wearable biomarkers in PD.
It is proposing a novel automated framework for imaging and clinical informatics and clinical beneftis are very important.
Innovative part should be more emphasized.
Establishing explainable digital phenotypes and supports the broader integration of wearable biomarkers into both clinical practice and research should be more emphasized.

**Weaknesses:**

Innovative part should be more emphasized.
Establishing explainable digital phenotypes and supports the broader integration of wearable biomarkers into both clinical practice and research should be more emphasized.